# National Level Cross-Sectional Study on Antibiotic Use in Children during the Pre- and Early COVID-19 Eras

**DOI:** 10.3390/antibiotics13030249

**Published:** 2024-03-09

**Authors:** Ji Young Park, Hyun Mi Kang

**Affiliations:** 1Department of Pediatrics, Korea University Ansan Hospital, Ansan 15355, Republic of Korea; ji8303@gmail.com; 2Department of Pediatrics, College of Medicine, Catholic University of Korea, Seoul 06591, Republic of Korea; 3Vaccine Bio Research Institute, College of Medicine, Catholic University of Korea, Seoul 06591, Republic of Korea

**Keywords:** antibiotics, children, coronavirus disease 2019

## Abstract

This study aimed to investigate national data for a quantitative evaluation of antibiotic usage in Korean children during the pre- and early COVID-19 period. This was a cross-sectional study from 2016 to 2021 of children <18 years, grouped by age (0, 1, 2–4, 5–11, and 12–17 years) and city/province. Systemic antibiotic prescriptions, days of administration, and population by age and region were collected. Days of therapy (DOT)/1000 pediatric inhabitant per day (PID) was used for antibiotic quantitative monitoring. A total of 257,088,265 antibiotic doses were prescribed to 170,309,944 children during the 6-year period. The highest DOT during the entire study period was observed in the 1-year age group, followed by the 2–4- and 0-year age groups. The highest DOT was observed in 2019, with 72.8 DOT/1000 PID in the 1-year age group, which fell to 34.7 DOT/1000 PID in 2020, however, DOT soon increased at similar rates to that in the pre-COVID-19 period. A higher DOT/1000 PID was observed for third-generation cephalosporins in 58.8% of the regions compared to beta-lactam/beta-lactamase inhibitors. To conclude, reductions in antibiotic use during the early COVID-19 pandemic period were not maintained. Further interventions are needed to decrease antibiotic overuse and misuse.

## 1. Introduction

Antimicrobial resistance has been increasing at alarming rates worldwide as antimicrobial overuse has become common in routine clinical practice. In children, antimicrobial prescribing rates by physicians in the United States were reported to have increased by 48% in office-based practice from 1980 to 1992 [1]. In Republic of Korea, the total consumption of antibiotics also increased from 21.68 defined daily doses (DDD) per 1000 inhabitants/day (DID) in 2008 to 23.12 DID in 2012 [2]. Furthermore, a previous study reported higher rates of urinary tract infections caused by extended spectrum beta-lactamase (ESBL) producers in regions with a higher amount of oral third generation cephalosporin prescriptions [3].

The World Health Organization (WHO) has presented a global action plan strongly urging countries to prepare countermeasures against indiscriminate antibiotic use. Likewise, many countries have undertaken interventions to endorse appropriate antibiotic use [4,5]. Since then, a decrease in antibiotic prescriptions for targeted diseases, such as respiratory tract infections, by off-based physicians has been reported [1,6]. In Republic of Korea, national cohort data showed a continuous downward trend in acute otitis medium-related broad-spectrum antibiotic use after adopting national action plans on antimicrobial resistance [7], demonstrating the success of multifaceted national efforts in certain disease categories. However, a decrease in the overall use of antimicrobials has not been clearly demonstrated at the population level.

In this study, we aimed to gather data to evaluate antimicrobial use on a national level over time to serve as a foothold for recognizing the need for the development of an antibiotic stewardship program (ASP) in pediatric care through conducting a quantitative evaluation of antibiotic use and commonly used antibiotic classes in Korean children. Furthermore, because the majority of antimicrobial overuse in children involves respiratory infections, we investigated the changes in antibiotic use during the pre- and early coronavirus disease 2019 (COVID-19) eras, when the incidence of many respiratory infections was reduced following strict pandemic control measures [8].

## 2. Results

### 2.1. Demographics and Analyses by Age Group and Year

A total 257,088,265 antibiotic prescriptions were made to 170,309,944 children <18 years of age during the 6-year period from January 2016 to December 2021 in Republic of Korea, with 96.32% of the prescriptions made in outpatient clinics, and 3.68% made in wards for admitted patients. The majority of the prescriptions for antibiotics were made in primary care clinics, accounting for 79.29%. Furthermore, oral antibiotics accounted for 96.57% of all prescriptions (Table 1).

The total antibiotic DOT/1000 pediatric inhabitants per day (PID) was analyzed for the age groups during each year of the study period. Children in the 1-year age group had the highest prescribed antibiotic DOT/1000 PID, followed by those in the 2–4-year and 0-year age groups. This pattern did not change significantly during the entire study period (age *p* < 0.001, year *p* = 0.460, and trend *p* = 0.491). In 2019, before the COVID-19 pandemic, the highest DOT/1000 PID was observed for all age groups, except the 5–11-year age group. During this peak, children in the 1-year age group were prescribed 72.8 DOT/1000 PID, which was the highest observed among all age groups during the entire study period. Then, in 2020, during the early COVID-19 period, this fell to 34.7 DOT/1000 PID (Figure 1).

### 2.2. Patterns of Prescribed Classes of Antibiotics by Age Group

In the 0- and 1-year age groups, the most commonly prescribed oral antibiotic classes were beta-lactam/beta-lactamase inhibitors, followed by third-generation cephalosporin and aminopenicillin. In the 2–4- and 5–11-year age groups, beta-lactam/beta-lactamase inhibitors were the most commonly prescribed antibiotic, followed by macrolides. In the 12–17-year age group, prescription of second-generation cephalosporin was most common, followed by prescription of beta-lactam/beta-lactamase inhibitors and macrolides (Figure 2a).

The most commonly prescribed intravenous antibiotic class in the 0-year age group was third-generation cephalosporin, followed by beta-lactam/beta-lactamase inhibitors, aminoglycosides, and penicillin. For the 1-, 2–4-, and 5–11-year age groups, beta-lactam/beta-lactamase inhibitors were the most commonly prescribed antibiotics, followed by third-generation cephalosporin, with almost the same percentage, and aminoglycosides. In the 12–17-year age group, aminoglycosides were the most commonly prescribed, followed by third- and first-generation cephalosporin (Figure 2b). 

### 2.3. Temporal Trends in Antibiotic Prescription

During the pre-COVID-19 period, a significant increasing trend was observed in the number of patients prescribed antibiotics, by approximately 16,307 patients per year (*p* < 0.001). However, when the COVID-19 pandemic was declared in March 2020, a significant decrease (by 2,220,054 patients) in the number of patients prescribed antibiotics was observed during the immediate COVID-19 pandemic period (*p* < 0.001). Nevertheless, as time passed, an increasing trend of approximately 18,028 patients per year was observed, which was similar to the increase observed during the pre-COVID-19 period (Figure 3).

Overall, from 2016 to 2017, a 39.8% (43.1 to 60.3 DOT/1000 PID) increase was observed, and in 2017 to 2018, 5.2% (60.3 to 63.4 DOT/1000 PID), and in 2018 to 2019, a 2.4% (63.4 to 64.9 DOT/1000 PID) increase was observed. However, from 2019 to 2020, a reduction of 50.5% (64.9 to 32.2 DOT/1000 PID) was observed, however, from 2020 to 20201, an increase of 5.3% (32.2 to 33.9 DOT/PID) was observed.

### 2.4. Analyses of Antibiotic Prescriptions by Region

For oral antibiotics, there were no significant differences in the prescription patterns of the three most commonly prescribed antibiotics, beta-lactam/beta-lactamase inhibitors, third-generation cephalosporin, and macrolides, in terms of their DOT/1000 PID (*p* = 0.999, Figure 4a). 

However, a significant difference was observed for the two most commonly prescribed intravenous antibiotics, beta-lactam/beta-lactamase inhibitors and third-generation cephalosporin, by region (*p* < 0.001), where some regions had an intravenous DOT/1000 PID more than four-fold greater than those of other regions. Moreover, a higher DOT/1000 PID was observed for third-generation cephalosporin in 10 of 17 regions (58.8%) (Figure 4b).

## 3. Discussion

Antibiotics are essential for treating focal and systemic bacterial infectious diseases. However, the prevalence of antimicrobial resistance correlates with the amount of antibiotic consumption, according to a summary combining Organization for Economic Co-operation and Development data and WHO data [9]. In this study, antibiotic use has been increasing every year, with variation in usage by age and region. Children in the 1-year age group had the highest prescribed antibiotic DOT/1000 PID, followed by those in the 2–4-year and 0-year age groups. Beta-lactam/beta-lactamase inhibitors were the most commonly prescribed antibiotics in children below 12 years of age, however, third-generation cephalosporins were prescribed at similar proportions, intravenously. During the pre-COVID-19 period, a significant increasing trend was observed in the number of patients prescribed antibiotics. However, a significant decrease was observed during the immediate COVID-19 pandemic period (*p* < 0.001). Nevertheless, as time passed, a similar increasing trend to the pre-COVID-19 period was observed.

The sustainability of global public health is threatened by antimicrobial resistance owing to the misuse and overuse of antibiotics. Unless global action is immediately initiated, the global population has the potential to be overcome by infectious diseases in a post-antibiotic era. In response to this crisis, the WHO assembly adopted the global action plan on antimicrobial resistance in May 2015 [10]. In line with this, the government of the Republic of Korea has also announced a national action plan on antimicrobial resistance every 5 years from August 2016. Nevertheless, the proportion of ESBL producers in healthy children within the community is increasing, as is the use of third-generation cephalosporins, as reported in previous studies [3,9]. In Korea, many hospitals do not yet have the division of ASP and do not have manpower allocation and data surveillance systems for monitoring antibiotic use. 

In order to define patterns of hospital antibiotic use, WHO developed the Access, Watch, and Reserve classification (AWaRe), where core access antibiotics included narrow spectrum antibiotics. WHO set an overall target that at least 60% of nationwide antibiotic use be from the Access group [11]. In a study 56 countries that took place between 2015 and 2017, the percentage of children prescribed Access antibiotics for lower respiratory tract infections varied from 10.3% in the Western Pacific region to 69.7% in Africa [12]. Although our study did not classify the antibiotics according to the AWaRe classification, the large proportion of antibiotics used included amoxicillin/clauvulanic acid, second-generation cephalosporin, and amikacin, which are all Access antibiotics. However, analyses by regions showed that Watch antibiotics, such as third-generation cephalosporins, were used at a much higher rate than amoxicillin/clauvulanic acid, belonging to Access antibiotics, warranting close monitoring. 

Quantitative monitoring of antibiotic use is difficult in children. In general, the DDD is a useful indicator for quantitative monitoring of antibiotic use because it does not require patient-level data. However, it is unsuitable for monitoring in children, where the antibiotic dose is determined according to body weight. In addition, as Korea has excellent medical access in terms of distance and cost, the follow-up visit interval is very short. Moreover, the visiting intervals may differ between tertiary hospitals and primary clinics. Therefore, it is not reasonable to use the number of antibiotic prescriptions as an indicator of antibiotic use monitoring for national [13] and institutional comparisons in Korea. Considering these issues in combination, the DOT is the most suitable indicator for monitoring antibiotic quantitative monitoring in children in Korea. The DOT requires information at the patient level, but as part of the project of the Korea National Antimicrobial Use Analysis System, the validation of HIRA public data with each institution’s data showed that the calculation of the DOT was also reasonable with public HIRA data. Therefore, in this study, we employed the DOT to quantitatively monitor antibiotic consumption in children. 

In accordance with the results of a previous study, we found that antibiotics were mostly prescribed in outpatient departments and primary care clinics [14]. In addition, the route of administration was almost always by the oral route. The DOT per person–day was the highest in children aged 1 year, followed by children aged 2–4 years and infants aged <1 year. This is possibly due to increases in exposure to many different transmissible pathogens compared to age 0; however, there is a lower level of immunity against the exposed pathogens compared to children in the older age groups. In the temporal trend analyses, antibiotic prescriptions sharply decreased in parallel with the declaration of the COVID-19 pandemic; however, since then, the same increasing rate of antibiotic prescriptions as the pre-COVID-19 period was observed. The prescribing patterns of oral and intravenous antibiotics also differed according to age, and there were differences in DOT per person–day by region and in patterns of preferred oral and intravenous antibiotics.

According to analyses of global antibiotic consumption by country, the change in DDDs per 1000 people days steadily increased between 2000 and 2015 in Korea. The antibiotic consumption rate in DDDs rapidly increased in upper–middle-income and low- and low–middle-income countries, while remaining nearly constant in high-income countries. However, Korea was one of the leading antibiotic consumers among the high-income countries in 2015 [15]. A nationwide study of the outpatient oral antimicrobial utilization pattern in Japanese children under 15 years old from 2013 to 2016 reported that, in 2013, a total 28.54 DOT/1000 PID were dispensed, whereas in 2016, it was 28.70 DOT/1000 PID. In our study, in 2016, the total antibiotic usage was 43.1 DOT/1000 PIDS; however, this increased, peaking at 64.9 DOT/1000 PID in 2019. When social distancing was applied during the early COVID-19 pandemic, the lowest antibiotic usage was still higher than the amount reported in Japan, which was 32.2 DOT/1000 PIDS [16]. 

In the US population, antibiotic use did not increase from 1999 to 2018; rather, the consumption of antibiotics has decreased according to a cross-sectional study using the National Health and Nutrition Examination Surveys. According to age, children aged 0–1 years had the highest rates of antibiotic prescriptions in the US, similar to our study results. However, in the US, decreases in antibiotic use were seen consistently throughout all age groups of children [17]. In Korea, antibiotic consumption in primary care centers consistently increased from 2010 to 2015 [18]. Although a previous study showed that the total amount of antibiotic use from 2008 to 2012 was stable [19], the majority of studies have shown that the total antibiotic consumption in Korea is steadily increasing [2,18,20]. Furthermore, only a few studies have investigated antibiotic consumption in populations including children [2,16,17]. The leading classes of antibiotics used in children were penicillin, third-generation cephalosporin, and macrolides [2,16]. However, all previous studies were analyzed with DDD or DID. Notably, children showed the highest DDD or DID, despite the small dosage of antibiotics prescribed by body weight; indeed, when calculated by DOT, there would be far more antibiotic prescriptions in children than in adults. Such data on DOT, a suitable indicator in children for antibiotic quantitative monitoring, are currently lacking in Korea. Thus, one of the main strengths of our study is the fact that it presents the first data on DOT in Korean children. Moreover, although the study period of most previous studies was from 2002 to 2015, we analyzed data from 2016 to 2021. Lastly, this study is the first to compare data from before the COVID-19 pandemic and in the COVID-19 era in Korea. 

Along with the declaration of a global COVID-19 pandemic, many countries, including Republic of Korea, immediately took actions to decrease the transmission rate by executing social distancing through school closures and mask wearing in public. Along with these measures, a sudden decline in respiratory and gastrointestinal viral infections, and to a lesser extent, a decline in bacterial infections, were observed [8,18]. In our study, the antibiotic prescription rates also correlated with the trend shown by the epidemiologic incidences of respiratory and gastrointestinal viral infections, where a sharp decline was observed in the immediate COVID-19 era. However, during the second year into the pandemic, with alleviated social distancing measures, such as returning to schools in children and alleviation of mask-wearing measures outdoors, the increase in the rate of antibiotic prescriptions returned to the same rate observed during the pre-COVID-19 era. Thus, this change in the antibiotic use pattern during the pre- and early COVID-19 period shown from our study, which is similar to the changes in epidemiology of viral infections, supports the presumption that the majority of antibiotic prescriptions are being given to infants and children that actually had a viral otitis media (OM) or upper respiratory infection to begin with. This shows that strong and urgent actions need to be taken to alter the antibiotic prescribing behaviors of physicians, and indiscriminate antibiotic prescribing for infants and children with OM or respiratory tract infections. Following updated guidelines that recommend stringent diagnostic criteria for acute OM can narrow down the group of children that actually have a pathogenic bacterial or combined viral etiology, thus decreasing indiscriminate antibiotic prescriptions [21]. 

This study has some limitations. First, we could not analyze all antibiotics. Moreover, there were masking data, when the data contained ≤3 patients, ≤3 companies producing the drugs, or the share of product from a specific company was ≥80%. Furthermore, the reimbursement antibiotics were excluded from the public data; therefore, these antibiotics were excluded from this study. Lastly, there is a limitation to the number of antibiotic-prescribed patients. Because HIRA data are based on monthly analysis, patients who are prescribed antibiotics from the end of the month to the beginning of the next month are counted as different patients. Therefore, to recover this limitation, the DOT was calculated as the denominator of the population, not the number of patients in this study. 

The Korea Disease Control and Prevention Agency has been steadily implementing policies to introduce and spread ASP activities to medical institutions. However, according to the fact-finding survey on ASP, the proportions with an incomplete understanding of ASP were 0% for tertiary general hospitals, 19.1% for general hospitals, and 50% for hospitals [22]. Moreover, according to this survey, the most difficult part in ASP operation is the lack of manpower. According to a previous study, national policy reduced inappropriate antibiotic prescribing [23]. These results show that it is necessary to improve ASP awareness in hospitals and primary care clinics, and highlight the need for national policies to perform ASP activities with appropriate personnel and monitoring systems. In particular, an antibiotic quantitative monitoring system in children, as the high antibiotic consumers, would be more important.

As consumption of antibiotics, such as penicillin and third-generation cephalosporin, increases in children, ESBL producers or methicillin-resistant organisms increase by selective pressure as well as acquiring resistance mechanisms. Thus, the need for prescriptions of broad-spectrum antibiotics, such as vancomycin and carbapenem, will inevitably increase, as will the emergence of vancomycin- and carbapenem-resistant organisms. Such multidrug-resistant organisms (MDROs) would require treatment by more broad-spectrum antibiotics, such as linezolid, colistin, or tigecycline; therefore, to reduce the emergence of MDROs, reducing antibiotic consumption in children is more important than infection control and monitoring the prescription of broad-spectrum antibiotics. 

## 4. Materials and Methods

This was a study of patients <18 years who were prescribed antibiotics from January 2016 to December 2021 in Republic of Korea across two time periods: during the pre-COVID-19 period (from January 2016 to February 2020) and during the early COVID-19 period (from March 2020 to December 2021). 

Antibiotics were defined as drugs corresponding to the J01 code of the Anatomical Therapeutic Chemical classification of the WHO. Antituberculosis, antiparasitic, and antiviral drugs were excluded, while intravenous and oral systemic antibiotics were analyzed. Topical antibiotics were excluded from the analysis. In Republic of Korea, all topical and systemic antibiotics need to be prescribed by licensed doctors and are unavailable as over-the-counter drugs. 

The health security system in Korea includes mandatory social health insurance and medical aid. The National Health Insurance system (NHIS) covers healthcare costs for all citizens. Thus, citizens are provided healthcare services by the healthcare institutions and make payments that have been reduced by the insurance rate. After providing healthcare services, the healthcare institutions are reimbursed by the NHIS for their services. The platform used to run this entire health insurance management is called Health Insurance Review and Assessment (HIRA) service (www.hira.or.kr, accessed on 31 July 2022). 

The patients were divided into the following age groups: (1) 0-year group, (2) 1-year group, (3) 2–4-year group, (4) 5–11-year group, (5) and 12–17-year group. The regions in Republic of Korea were divided into 17 regions as follows: eight cities, including the capital city of Seoul, the self-governing city of Sejong, and the metropolitan cities, Busan, Daegu, Daejeon, Gwangju, Incheon, and Ulsan; and nine provinces, including the self-governing provinces Gangwon and Jeju, North Chungcheong, South Chungcheong, Gyeonggi, North Gyeongsang, South Gyeongsang, North Jeolla, and South Jeolla.

The number of patients prescribed systemic antibiotics, prescriptions, and days of administration were collected from the HIRA service, through the Korean government’s public big data repository (www.data.go.kr, accessed on 31 July 2022). Due to the nature of public data, it was difficult to collect an accurate number of patients due to monthly analysis. Therefore, in this study, the number of days of therapy (DOT) per 1000 PID was evaluated with the population from the age group as the denominator (=DOT × 1000)/(pediatric inhabitants × 365). The number of populations by age group and region was collected from the national statistical portal (Korean Statistical Information Service, https://kosis.kr, accessed on 31 July 2022), which uses personal identification numbers to count the population (Table 2).

A total of 247 antibiotics were included in the analysis and classified into 19 classes: oral antibiotics, including aminopenicillin, beta-lactam/beta-lactamase inhibitor, first-generation cephalosporin, second-generation cephalosporin, third-generation cephalosporin, macrolide, quinolone, and tetracycline; and intravenous antibiotics, including penicillin, beta-lactam/beta-lactamase inhibitor, first-generation cephalosporin, second-generation cephalosporin, third-generation cephalosporin, fourth-generation cephalosporin, macrolide, quinolone, aminoglycosides, carbapenem, and glycopeptide. 

This study was approved by the Institutional Review Board of Seoul St. Mary’s Hospital (IRB No. KC23ZASI0678). All statistical analyses were performed using R version 3.2.1. (R Foundation for statistical computing, Vienna, Austria). Categorical variables were analyzed using Fisher’s exact test, and continuous variables were analyzed using the Mann–Whitney U test. Linear regression analyses were used to investigate the overall temporal trend. Interrupted time series analysis was performed to observe a difference in antibiotic use before and during the COVID-19 pandemic, using March 2020 as the time point. All statistical tests were two tailed, and a *p*-value < 0.05 was considered significant.

## Figures and Tables

**Figure 1 antibiotics-13-00249-f001:**
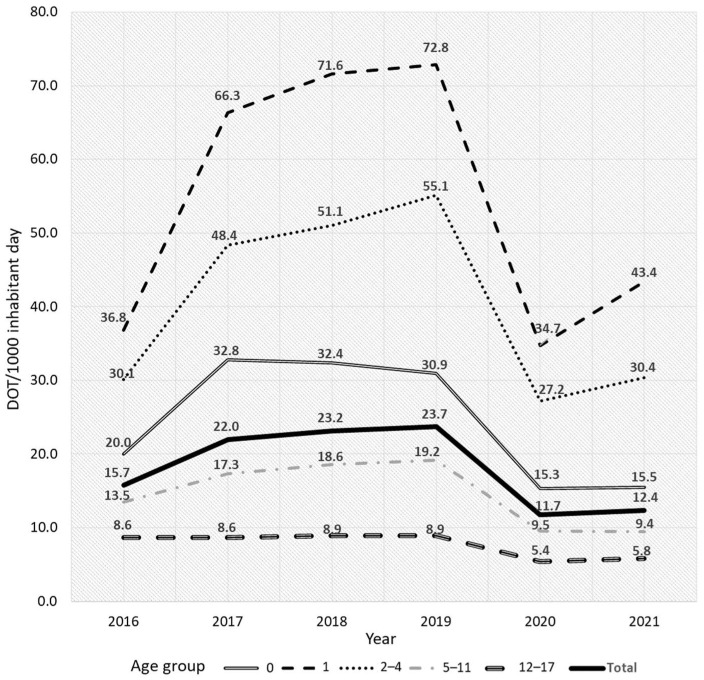
Antibiotic days of therapy (DOT) per 1000 PID by age group and year. Total antibiotic DOT/1000 PID showed the highest prescribed antibiotic DOT/1000 PID in the 1-year age group followed by 2–4-year and 0-year age groups. In 2020, during the early COVID-19 period, a decrease in antibiotic DOT was observed throughout all age groups. PID, pediatric inhabitants per day.

**Figure 2 antibiotics-13-00249-f002:**
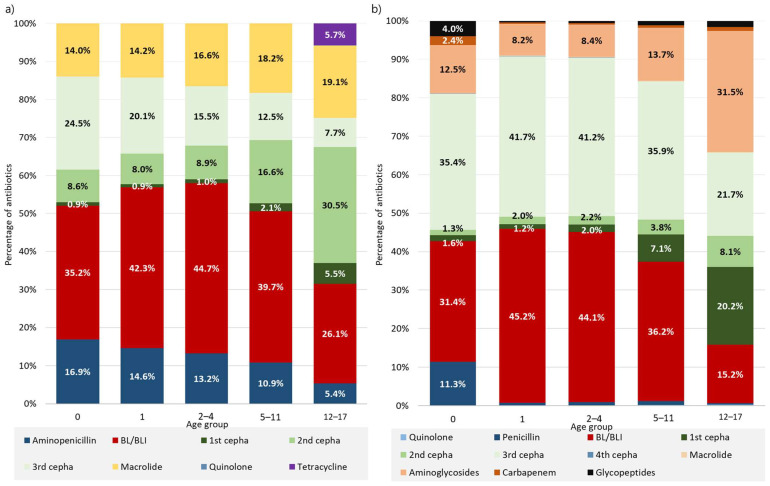
Prescribed (**a**) oral and (**b**) intravenous antibiotic classes by age group. (**a**) The most commonly prescribed oral antibiotic class was beta-lactam/beta-lactamase inhibitors in all age groups, except in the 12–17-year age group, where second-generation cephalosporin was the most common. (**b**) The most commonly prescribed intravenous antibiotic class in the 0-year age group was third-generation cephalosporin. In 1-, 2–4-, and 5–11-year age groups, beta-lactam/beta-lactamase inhibitors were the most commonly prescribed antibiotics, and in the 12–17-year age group, aminoglycoside was the most commonly prescribed antibiotic class.

**Figure 3 antibiotics-13-00249-f003:**
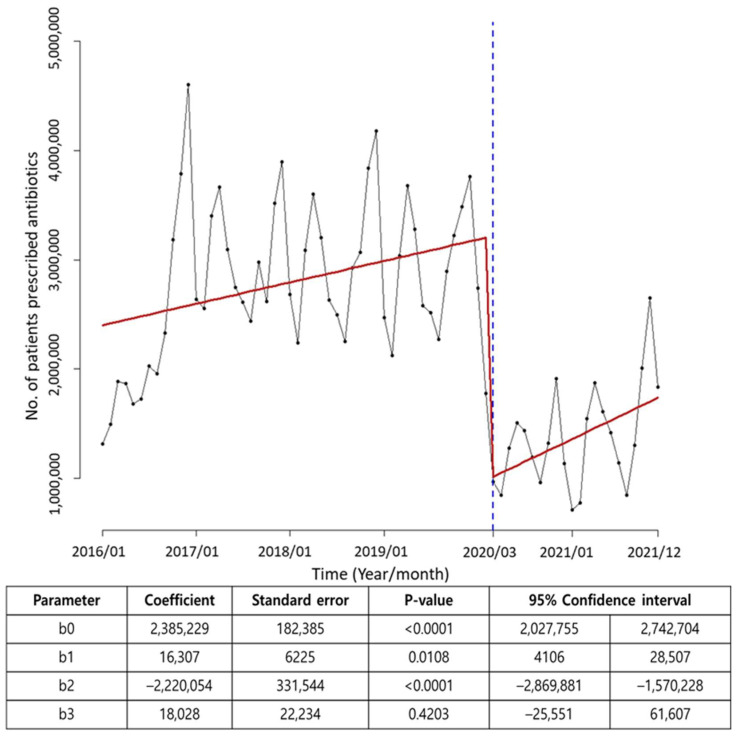
Interrupted time series analysis using March 2020 (COVID-19 pandemic declaration) as a time point. During the pre-COVID-19 period, a significant increasing trend was observed in the number of patients prescribed antibiotics. However, a significant decrease was observed during the immediate COVID-19 pandemic period (*p* < 0.001). Nevertheless, as time passed, a similar increasing trend to the pre-COVID-19 period was observed. Blue dash line represents the declaration of the COVID-19 pandemic.

**Figure 4 antibiotics-13-00249-f004:**
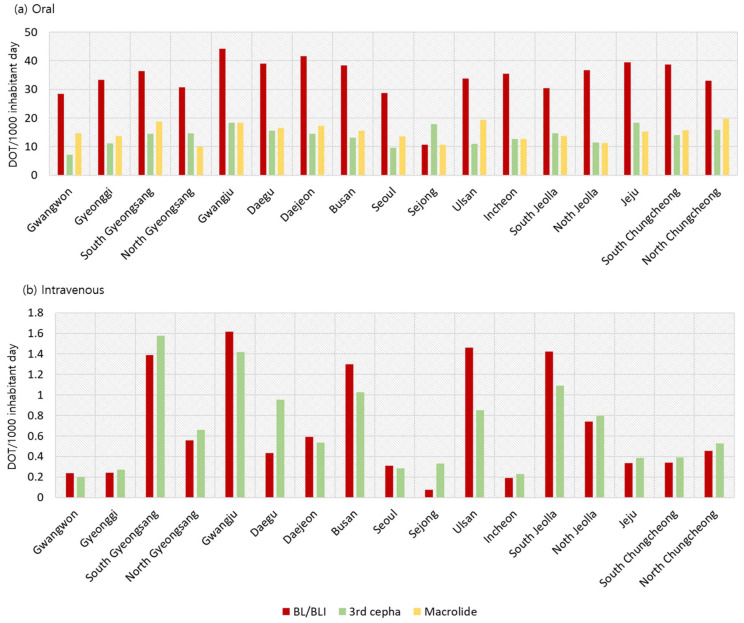
(**a**) Oral and (**b**) intravenous antibiotic days of therapy (DOT) per 1000 PID by region. (**a**) For oral antibiotics, there were no significant differences in the prescription patterns of the three most commonly prescribed antibiotics. (**b**) A significant difference was observed for the two most commonly prescribed intravenous antibiotics, beta-lactam/beta-lactamase inhibitors and third-generation cephalosporin, by region. PID, pediatric inhabitants per day.

**Table 1 antibiotics-13-00249-t001:** Demographics and clinical data included in this study.

	No. of Patients		*p*	No. of Prescriptions		*p*
	*N* = 170,309,944	%		*N* = 257,088,265	%	
Prescribed place			<0.001			<0.001
Outpatient department	161,087,148	94.58		247,633,761	96.32	
Ward	9,222,796	5.42		9,454,504	3.68	
Type of medical care center			<0.001			<0.001
Tertiary general hospital	2,465,394	1.45		2,716,220	1.06	
General hospital	9,097,375	5.34		11,510,181	4.48	
Hospital	25,487,538	14.97		36,939,583	14.37	
Primary care clinic	131,411,559	77.16		203,832,791	79.29	
Others	1,848,078	1.09		2,089,490	0.81	
Administered route			<0.001			<0.001
Oral	162,634,775	95.49		248,272,470	96.57	
Intravenous	7,675,169	4.51		8,815,795	3.43	
Age group						
0	8,190,624	4.81		13,409,391	5.22	
1	20,130,364	11.82		34,373,178	13.37	
2–4	53,420,071	31.37		86,923,974	33.81	
5–11	57,894,436	33.99		83,205,024	32.36	
12–17	30,674,449	18.01		39,176,698	15.24	

**Table 2 antibiotics-13-00249-t002:** Number of children below 18 years according to yearly consensus and number of newborns per year.

Age Group (Years Old)	2016	2017	2018	2019	2020	2021
0	393,674	345,786	317,685	295,132	265,087	253,946
1	441,720	409,814	361,625	330,970	304,651	274,633
2–4	1,368,877	1,323,515	1,294,934	1,219,020	1,107,285	1,001,890
5–11	3,248,187	3,302,078	3,293,789	3,239,207	3,215,416	3,180,414
12–17	3,283,593	3,099,254	2,908,302	2,844,578	2,818,507	2,773,061
Total	8,736,051	8,480,447	8,176,335	7,928,907	7,710,946	7,483,944
Newborns	406,243	357,771	326,822	302,676	272,337	260,562

Data from Korean Statistical Information Service (https://kosis.kr, accessed on 31 July 2022).

## Data Availability

Data are available upon request to the corresponding author.

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
