# Peer review of "National Level Cross-Sectional Study on Antibiotic Use in Children during the Pre- and Early COVID-19 Eras"

_antibiotics, 2024, doi:10.3390/antibiotics13030249_

Round 1

Reviewer 1 Report

Comments and Suggestions for Authors

The aim of the study was to investigate national data of antibiotic usage in South Korea during pre -and early COVID-19 pandemic.

My comments:

-DDD/PID (pediatric inhabitants per day) is probably more correct

-Table 1.I miss in the Table 1. number of patients and number of prescriptions  according to age group 0,1,2-4, 5-11-12-17 and total

-It looks that Korea does not use ATC/DDDs  system.They included in the analysis 19 classes -TMP/SMX, lincosamides and other antibiotics commonly used in children in outpatients are not analyzed. Total use of J01 is probably not correct.

-COVID 19 ; lifted measures  are not clearly presented

-What is the number of children <18 years according to yearly census and number of newborns/year

-Have all children health insurance

-does exist OTC or are antibiotics purchased only by doctors ( pediatricians, GPs)prescriptions?

-What was yearly increase of prescribed antibiotics expressed in % in pre COVID-19 period?

- What are factors influenced high prescriptions in children of 1 year of age.

- Figure 2. classes of oral antibiotics represent 100% of them.Were other classes not included?

-reimbursement antibiotics were not included. How the parents can get antibiotics? Do they pay?

Discussion is very limited.The authors should include in the discussion also the EU countries and Japan. The comparison of number of prescriptions per 1000 children per year and DOT/PID would be beneficial.  During COVID-19 substantial decline of antibiotic consumption in children and in the community was observed for most countries. The characteristic of South Korea is high usage of broad spectrum antibiotics (AWaRe) in children. These should be mentioned ( Hsia Y et al Lancet Infect Dis 2018). This study only confirms such data. 

The added value of the manuscript is very limited. It should be improved.

Author Response

Dear Reviewer,

Thank you very much for taking the time to review this manuscript. We have replied to the comments made, and revised the manuscript accordingly.

Please refer to the attached files, where I have written the response to reviews. 

Sincerely, 

Hyun Mi Kang

Reviewer 2 Report

Comments and Suggestions for Authors

Thank you for this opportunity to review this well written manuscript detailing your analysis of antibiotic prescribing in Korean children between 2016 and 2021. The most fascinating aspect of your results is the significant reduction in prescribing during the Covid-19 lockdown. You explain this very well in the Discussion, and of course similar patterns were observed in other countries as a result of social distancing and masking. However, the second sentence of paragraph 6 in the Discussion is particularly interesting: "Along with these measures, a sudden decline in respiratory and GI viral infections, and to a lesser extent, a decline in bacterial infections was observed." I would like to see this explored and discussed further:

1. You are rightly building the argument that antibiotic prescribing quickly returned to the inappropriate levels observed pre-pandemic. Does the fact that there was a lesser decline in bacterial, rather than viral infections, during the lockdown support in any way the presumption that the majority of antibiotic prescriptions are being given to infants and children that actually have a viral otitis media (OM) or upper respiratory infection to begin with?

2. One of the key arguments about indiscriminate antibiotic prescribing has always revolved around outpatient prescribing for infants and young children with OM or upper respiratory tract infection. I would suggest referring to the current treatment guidelines, which includes a review of the most likely pathogens by age group (predominantly viral in infants and young children). 

Author Response

(The authors gave the same response as above.)

Round 2

Reviewer 1 Report

Comments and Suggestions for Authors

The authors answered to comments.The paper is improved and more clear.